# *Galleria mellonella* immune melanization is fungicidal during infection

Daniel F. Q. Smith [1], Quigly Dragotakes [1], Madhura Kulkarni[1], J. Marie Hardwick [1] &
Arturo Casadevall [1 ✉]

A key component of the insect immune response is melanin production, including within nodules, or aggregations of immune cells surrounding microbes. Melanization produces oxidative and toxic intermediates that limit microbial infections. However, a direct fungicidal role of melanin during infection has not been demonstrated. We previously reported that the fungus *Cryptococcus neoformans* is encapsulated with melanin within nodules of *Galleria mellonella* hosts. Here we developed techniques to study melanin's role during *C. neoformans* infection in *G. mellonella*. We provided evidence that in vivo melanin-encapsulation was fungicidal. To further study immune melanization, we applied tissue-clearing techniques to visualize melanized nodules in situ throughout the larvae. Further, we developed a time-lapse microscopy protocol to visualize the melanization kinetics in extracted hemolymph following fungal exposure. Using this technique, we found that cryptococcal melanin and laccase enhance immune melanization. We extended this approach to study the fungal pathogens *Candida albicans* and *Candida auris*. We find that the yeast morphologies of these fungi elicited robust melanization responses, while hyphal and pseudohyphal morphologies were melanin-evasive. Approximately 23% of melanin-encapsulated *C. albicans* yeast can survive and breakthrough the encapsulation. Overall, our results provide direct evidence that immune melanization functions as a direct antifungal mechanism in *G. mellonella*.

[1] W. Harry Feinstone Department of Molecular Microbiology and Immunology, The Johns Hopkins Bloomberg School of Public Health, Baltimore, MD 21205, USA. ✉email: acasade1@jh.edu

nsects occupy essential niches in global ecosystems, including many that directly affect human health and survival[1]. In addition, insects serve as powerful model systems for infectious disease research, and help to reduce reliance on vertebrates recommended by "3R"—Replace, Reduce, and Refine—programs[2]. Insects are also targeted by environmental pathogens and have evolved complex immune mechanisms that partially overlap with mammalian innate immunity[3]. Understanding the dynamics of insect-pathogen interactions and the factors involved is vital to both ensure ecosystem stability and establish invertebrate immunological models in research.

Fungi are an important class of pathogens for insects, and emerging fungal pathogens are predicted to become bigger threats to human health and agriculture in the coming years[4,5]. Consequently, studying host-fungal interactions using insect models is important and timely. Although insects do not produce antibodies or other mammalian-like adaptive immune responses, the antifungal immune defenses of insects involve cell-mediated and humoral innate immune processes[3]. Hemocytes, the immune cells of invertebrates which circulate in the hemolymph, have roles comparable to macrophages and neutrophils in mammals. Hemocytes are responsible for clearance of fungi via phagocytosis, release of extracellular damaging reactive oxygen species (ROS) and inflammatory molecules, and the creation of granuloma-like structures through a process called nodulation[6]. During nodulation, hemocytes surround the microbe and form an aggregate of insect cells, within which, clotting factors, immune enzymes, and immune complexes are released and activated[6–8]. These structures immobilize the fungus and lead to death. In Lepidopteran species, the production of prostaglandins by the plasmatocyte subset of hemocytes cause the lysis of other hemocytes called oenocytoids, resulting in the release of antimicrobial peptides, signaling molecules, and enzymes important to immune function[9–11]. One class of host enzymes that are often released and activated during oenocytoid lysis and nodulation are phenoloxidases (PO)[9,11]. POs are enzymes responsible for converting catecholamines in the hemolymph into DOPA melanin[12]. DOPA melanins, and melanins more broadly, are black-brown pigments contribute to insect immune defense and wound repair[13]. Melanization produces oxidative species and cytotoxic intermediates that are hypothesized to result in the death of the microbe[12,14]. Additionally, melanin may act as a physical barrier, restricting gas exchange and nutrient uptake, and thus prevent fungal replication and dissemination to other tissues[15]. At this time, in vitro evidence strongly links PO activity and resulting melanin intermediates with killing of fungi, bacteria, and viruses[16–18], but comparable direct evidence for the microbicidal effect of POs and their toxic intermediates in vivo during insect infections is challenging to measure directly. Consequently, obtaining direct evidence that the process of melanization is fungicidal in vivo is important for establishing insect melanin as an important antifungal mechanism.

Larvae of the wax moth Galleria mellonella are commonly used as a model organism for studying fungal pathogenesis[3]. G. mellonella larvae are readily available in large numbers at low cost. Their larger size (2–3 cm) relative to other model insects such as Drosophila melanogaster makes them amenable to experimental approaches requiring larger volumes of hemolymph, insect hemocytes, and soluble immune factors. The study of G. mellonella hemolymph can prove valuable for understanding insect immune responses to infection. G. mellonella are also commonly used as a model for studying mammalian pathogens, including human pathogenic fungi Cryptococcus neoformans and Candida albicans[19–21]. While G. mellonella is a model for mammalian fungal infections because of similarities between the G. mellonella immune responses and the mammalian innate immune responses[3], a more thorough understanding of the insect immune response is needed to fully benefit from studying host-microbe interactions in G. mellonella.

The differences between mammalian versus insect hosts also provide important insights into host-microbe interactions and mechanisms of fungal virulence factors[3,19,20,22,23]. For example, laccase, a fungal enzyme that oxidizes mammalian and insect catecholamines, is an important virulence factor in both hosts but, seemingly by distinct and diverse mechanisms[19,24,25]. In insects, fungal laccase appears to oxidize and deplete host catecholamines required for encapsulating the fungus in melanin, thus weakening the host immune response. Fungal laccases also help detoxify reactive oxygen species that form during insect immune processes[24]. In contrast, during mammalian infection, fungal laccase enhances production of fungal melanin to evade key mammalian immune defenses[26]. Thus, fungal melanins increase virulence in mammals, but decrease virulence in G. mellonella[27–29]. These seemingly different and opposite roles in which fungal melanins interact with mammalian and insect hosts is unexplored and as of now unexplained in literature.

In this study, we demonstrate the first direct evidence, to the best of our understanding, that the melanin-based immune response in vivo is fungicidal against C. neoformans. Fungal death was visualized using an endogenously expressed GFP-based fungal viability assay. A series of in situ, ex vivo, and in vitro methods to observe and study the melanin-based immune response of G. mellonella larvae. For in situ experiments, a previously published tissue-clearing protocol was modified to visualize tissue-specificity of melanized nodules within an intact larva. We have also developed time-lapse microscopy method for observing the melanin-based immune system in vitro. We applied this method to quantify the melanization kinetics during in vitro fungal infection, which improved our understanding of how fungal components, such as laccase and melanin, interact with insect melanization. We gained insight into how Candida albicans and Candida auris activate and evade the melanin-based immune response through morphological switching. Overall, our findings strongly suggest that melanization has direct antimicrobial activity in vivo in the insect immune system.

## Results

**Galleria mellonella kill C. neoformans through melanin encapsulation in nodules.** Our group previously identified that C. neoformans infection activates the insect-melanin-based immune response and encapsulation response in G. mellonella[30], where hemocytes aggregate and produce melanin around the fungus (Fig. 1a). To evaluate whether insect-melanin encapsulation kills C. neoformans, we assessed fungal viability during infection of G. mellonella using a GFP-expressing strain of C. neoformans, which expresses GFP under an actin promotor and shows loss of fluorescence signal upon cell death. A similar assay using the GFP signal as a proxy for parasite viability within mosquitoes was previously described[31]. The GFP-expressing strain was validated as a reporter for fungal viability C. neoformans using the standard dead cell stain propidium iodide (PI) (Supplementary Fig. 1a, b). In these experiments, propidium iodide staining was nearly mutually exclusive with GFP fluorescence in untreated fungal cells, and GFP fluorescence was lost following fungal death when treated with high temperatures (Fig. 1b, Supplementary Fig. 1a, b).

Following infection of G. mellonella larvae, we found fewer GFP-positive melanin-encapsulated fungal cells within nodules of infected larvae compared to GFP-positive non-melanin-encapsulated fungal cells at both room temperature and at 30 °C at 24 and 72 h post-infection (Fig. 1c, d). Using the counts

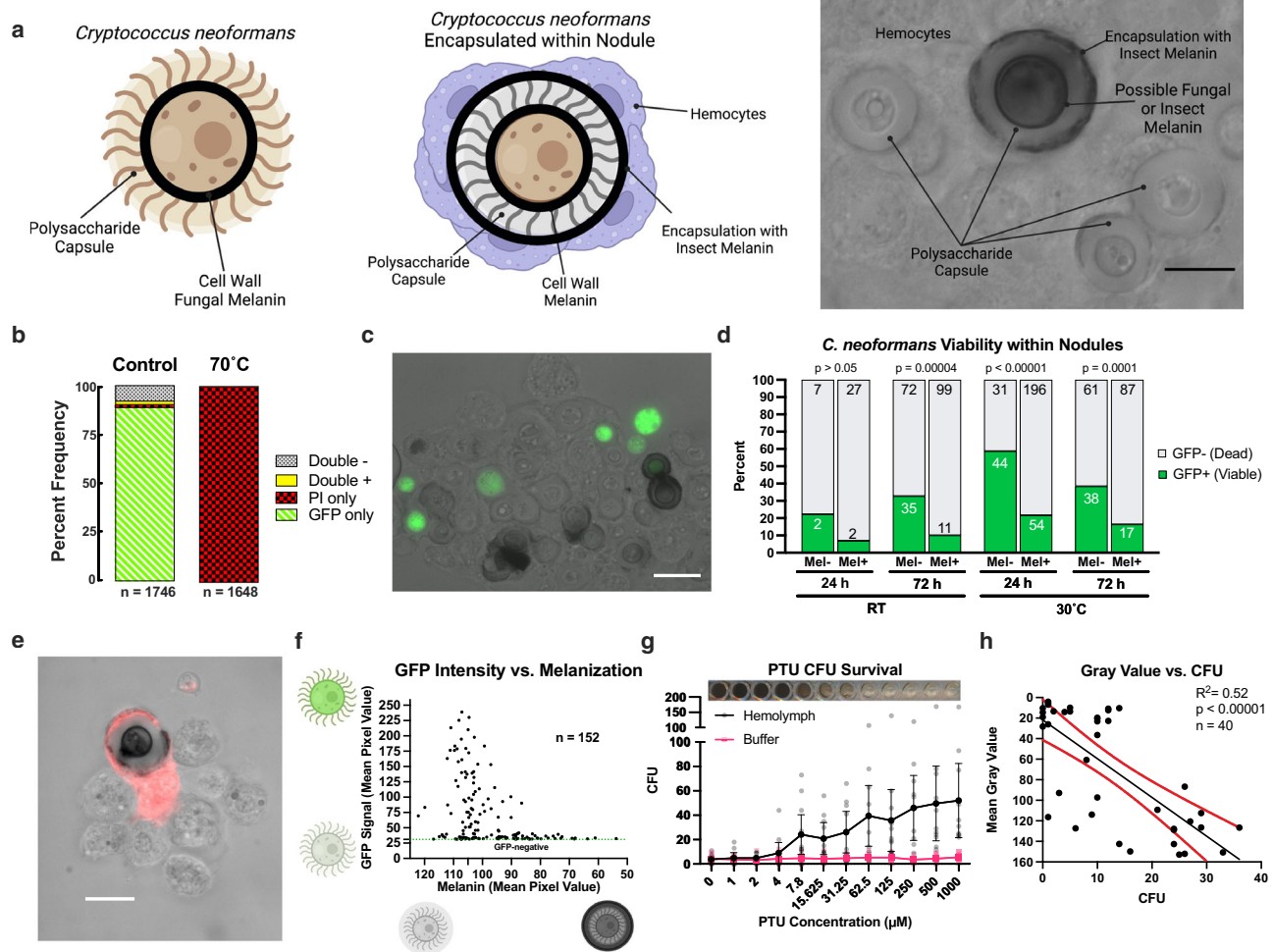

**Fig. 1 The melanin-based immune response is implicated in killing *C. neoformans* during infection. a** During infection of *G. mellonella*, melanized nodules are formed within the hemolymph, which encapsulate *C. neoformans*. Melanin pigment produced by the fungus, is located typically within the cell wall, while the melanin produced by the insect during encapsulation within nodules is found around and within the fungal polysaccharide capsule. **b** GFP fluorescence is lost in heat-killed *C. neoformans* cells expressing GFP under an actin promoter. Data represents counts of 1640–1750 single cells from two separate biological replicates. **c** Using a GFP-expressing strain of *C. neoformans*, we can visualize cell viability within the nodule, with living fungi fluorescing green and dead cells having no signal. **d** Melanin-encapsulated fungi are statistically less likely to have a GFP-positive signal at 24 h after infection at 30 °C, and at 72 h post-infection at room temperature and 30 °C (Chi-squared test, [$n = 217$ cells] $p < 0.00001$, [$n = 325$ cells] $p = 0.00004$, and [$n = 203$ cells] $p = 0.0001$, respectively) indicating that melanin-encapsulation is associated with fungal death. Numbers within bars represent the number of fungi enumerated in that condition. **e** In GFP-positive cells, the brighter fluorescence signal correlates with little to no melanization, with no cells that have strong GFP signals and large amounts of melanin encapsulation ($n = 152$ independent cells). Each point represents data from one GFP-positive fungus within nodules from three biological replicates. **f** Propidium Iodide viability staining does not appear to penetrate the inner space of the nodule where the fungus is located, but the staining does show that the hemocytes surrounding the fungus are not viable. **g** Colony forming units (CFUs) correlate directly to melanin inhibition using PTU (inset, representative image of the 96-well plate) in vitro, which indicates that melanin has a role in controlling growth of *C. neoformans*. Line represents the mean and error bars indicate 95% Confidence Interval of four independent biological replicates with 10–12 total technical replicates. **h** CFUs also inversely correlated to the Mean Gray Value of these wells, meaning that the darker wells with more melanization had higher CFUs recovered compared to the melanin-inhibited wells (Simple Linear Regression, $n = 40$ biologically independent wells, $p < 0.00001$, F = 40.59). Each data point indicates measurements from individual wells from across three biological replicates, red dotted lines represent the 95% confidence interval of the linear regression. **a–c, f** Data shown is representative of three biological replicates. All experiments performed in biological triplicate, with **g** representing 5 biological replicates. Scale bars represent 10 μm.

of GFP positivity of fungi within nodules, he non-melanin-encapsulated fungi had a survival rate of approximately 22, 33, 59, and 38% for the room temperature 24 and 72 h and the 30 °C at 24 and 72 h conditions, respectively (Fig. 1d). The melanin-encapsulated fungi had a survival rate of approximately 6, 10, 21, and 16% for the room temperature 24 and 72 h and the 30 °C at 24 and 72 h conditions, respectively (Fig. 1d). These data show that melanin-encapsulation was thus associated with approximately 21, 34, 92, and 35% greater amount of fungal death at the temperatures and times tested. This indicates that melanization

contributes to a significant amount of fungal death during infection but is not the only nor predominant mechanism by which fungal cells are killed. Additional causes of fungal death in this model may include antimicrobial peptides and respiratory bursts, as previously described[3]. Melanin produced by *C. neoformans* in culture did not quench or obscure the GFP fluorescence, as determined by imaging melanized versus non-melanized *C. neoformans* H99-GFP (Supplementary Fig. 1c, d). Therefore, these results suggests that the immune melanization reaction resulted in fewer GFP-positive cells, which indicated that

melanization kills *C. neoformans* in vivo during infections of *G. mellonella*.

Within nodules of encapsulated GFP-positive *C. neoformans* extracted from the hemolymph of infected *G. mellonella*, we measured the degree of immune melanin intensity and GFP fluorescence intensity. We found that the yeast with weaker GFP signal were associated with greater melanin encapsulation, in contrast to the population of brightly fluorescent cells that were not encapsulated with as much, or any, melanin (Fig. 1e), thus identifying an inverse correlation between degree of melanin encapsulation and fluorescence within GFP-positive cells. The occurrence of faint signal in some cells may suggest that cells are undergoing cell death or have recently been killed.

We attempted to use the standard dye propidium iodide (PI) to validate that GFP-negative cells within nodules are dead. Surprisingly, PI staining did not stain nodule-encapsulated yeast cells, although there was staining in some of the hemocytes that surrounded the yeast cells, and the external periphery of the nodules (Fig. 1f). The absence of PI staining of the fungi is likely due to inability of the dye to penetrate the nodule fully.

To further evaluate whether insect melanin promoted death of *C. neoformans*, fungal cells were incubated with whole extracted *G. mellonella* hemolymph in the presence of increasing concentrations of a phenoloxidase-specific competitive inhibitor[32], phenylthiourea (PTU). This generated a range of insect-melanin inhibiting conditions. After 24 h, a small aliquot of the hemolymph-fungal mixture was plated on nutrient rich agar to assess fungal growth. Incubation of fungus melanization-inhibited hemolymph in vitro drastically enhanced fungal growth based on CFUs, while fungal growth was nearly abolished in melanin-competent hemolymph (Fig. 1g). The number of *C. neoformans* CFUs following 24 h incubation in the presence of hemocytes and PTU melanin inhibitor was directly proportional to the concentration of PTU (Fig. 1g), and thus inversely proportional to the degree of melanization (Fig. 1g, inset). PTU-mediated suppression of melanization was verified by quantifying melanin intensity by mean gray value of the wells of the 96-well plate (Fig. 1g, inset). The mean gray values of the wells showed that greater suppression of melanin production by PTU treatment correlated with increasing fungal growth based on CFUs (Fig. 1h). This result strongly suggests that immune melanization inhibits the growth of *C. neoformans* in vitro. Control conditions with buffer and PTU alone showed that incubation with PTU did not have a negative inhibitory effect on fungal growth, nor did it account for increased fungal growth in the high concentrations.

**Development of a time-lapse microscopy method for studying hemocyte-fungal interactions and the melanization response**. Since melanin encapsulation plays an important role in insect immunity and killing *C. neoformans* during infection, we sought to develop techniques to better study the melanization reaction and melanin-based host-fungal interactions. To record the kinetics of the hemolymph melanization response, we developed a protocol to extract insect hemocytes and monitor fungal interaction in vitro. (Fig. 2a). This method allowed for visualizing and quantifying the rate and magnitude of the anti-cryptococcal immune melanization response and was extended to study insect infection using different species of fungi, gene-deletion mutants, or application of isolated virulence factors (Supplementary Movie 1, 2). Using particle analysis in recorded frames, we quantified the area covered by melanization in minute intervals to determine the rate of hemolymph melanization (Fig. 2b–d). We observed variation in the extent of melanization between different experiments, likely due to variability in the fungi in the field of view, and biological variability from the non-isogenic *G.*

*mellonella* larvae. To overcome the risk of interpreting variability-derived artifacts as results, each experiment was performed with a corresponding control (i.e., parental strain and mutant experiments were performed at same time, using the same stock of hemolymph and the same pool of extracted hemocytes). Despite the variation, the method was helpful in showing how morphological differences or fungal components interact with the melanin immune response, as well as comparing magnitude of melanization and speed of melanization activation between two groups.

By analyzing the melanization kinetics between different mutant and wild-type strains, we can determine how the mutant gene of interest affects fungal interactions with the melanization immune response. In *C. neoformans*, the laccase enzyme Lac1 is a major virulence factor, as it is responsible for the oxidation of catecholamines, and thus responsible for melanization[33]. It has also been suggested that laccase plays roles in virulence beyond melanization within mammalian hosts[34]. In a previous study the *lac1Δ* mutant showed attenuated virulence in *G. mellonella*.[19] We found that there was a reduced rate and magnitude of hemolymph melanization following infection with *lac1Δ* mutant compared to the wild-type parental strain (Fig. 2e). While the overall magnitude of melanization varied between replicates, the ratio of insect melanization that occurred in the *lac1Δ* versus H99 (WT) was consistently lower (Fig. 2f).

Many fungi, including most human pathogens, produce either DOPA or DHN melanin in their cell walls, which may enable them to persist within mammalian hosts and avoid destruction from oxidative stress and antimicrobial agents[26]. However, several studies have shown that melanized fungi are less virulent in *G. mellonella* compared to their non-melanized or albino mutant counterparts[27,28,35]. We hypothesized that the fungal melanin could act as a pathogen or damage-associated molecular pattern, resulting in enhanced activation of the melanization immune reaction. We found that isolated DOPA melanin produced by *C. neoformans* (termed melanin ghost, which are isolated following boiling 6 N hydrochloric acid[36]) activated immune melanization, both with and without hemocytes present compared to heat-killed *C. neoformans* (Fig. 2g, Supplementary Movie 3–5). Melanin ghosts contain trace amounts of fungal cell wall components that may activate the melanization immune response[37,38], therefore non-melanized heat-killed cells were used as a control for the cell wall components that would be present in small amounts. The heat-killed cells did not activate a melanization response like the fungal melanins. Hence, it appears that isolated fungal melanin is specifically recognized and activates the phenoloxidase cascade. Further, the isolated melanin ghosts are aggregated by the hemocytes throughout the course of the time-lapse microscopy, even when immune melanization is not activated (Supplementary Movie 6).

We used these techniques to compare the immune melanization between *C. neoformans* and *C. albicans*, the latter of which is known to trigger robust melanization of the hemolymph[39]. Following infection, *C. albicans* activated the melanization response faster (beginning as early as 15 min) and to a significantly greater extent than did *C. neoformans* (Fig. 2h). These data showing large amounts of melanization induced by *C. albicans* corresponds to the levels of melanization previously reported that occurs during *G. mellonella* infection with *C. albicans* versus *C. neoformans* and further validates the developed methodology[39].

**Evaluating the melanin-based immune response of *G. mellonella* using tissue clearing**. Tissue clearing is a technique that allows for visualization of structures deep within an organism or

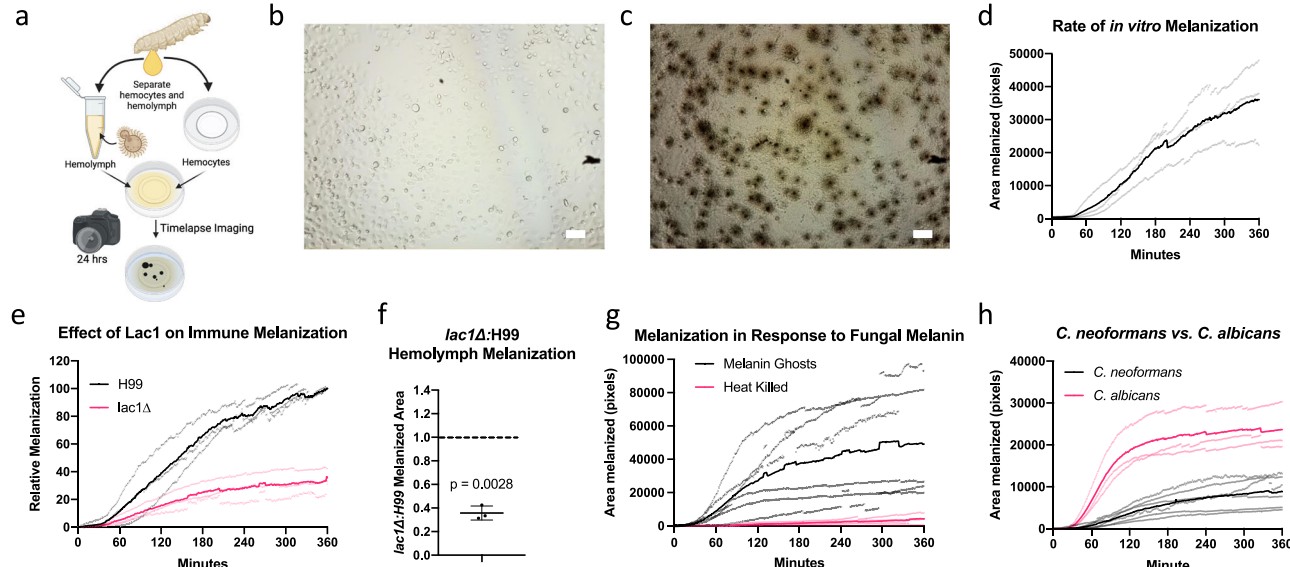

**Fig. 2 Using in vitro time-lapse microscopy to visualize the melanin-based immune response. a** Using a developed time-lapse microscopy protocol, we were able to record the dynamics of hemolymph melanization in response to C. neoformans from 0 (**b**) up to 24 h (**c**). We can then use particle measurement software to visualize and quantify the melanization reactions over time (**d**), representative of three biological replicates. This method can be used to gain insight into how different virulence factors, such as laccase, influence the melanization response, where we see that the laccase knockout mutant causes less of a melanization response without much change in time until onset of melanization (**e**), represented by three biological replicates, and normalized to the wild-type control melanization area at 360 min. **f** Overall, the lac1Δ triggers ~40% as much of a melanization immune response compared to the WT (one sample t-test, n = 3 biologically independent movies, p = 0.0028, t = 18.78, df = 2). **g** Isolated fungal melanins are associated with activation of the insect-melanin-based immune response, whereas the heat-killed acapsular cryptococcal cells are not, as represented by 3–6 biological replicates. **h** Additionally, we can compare the activation of the melanin-based immune response from different fungal species such as C. neoformans and C. albicans, which strongly activates the melanization immune response, represented by 3–6 biological replicates. **b–d**, **g** are representative images and graphs from three biological replicates. Error bars indicate Standard Deviation. **d**, **e**, **g**, **h** Solid lines indicate mean melanization, and light colored points indicate individual replicate data points. Scale bars represent 50 μm.

tissue sample, without significant disruption of the native tissue anatomy. A previously reported protocol[40] was adapted to visualize the anatomical localization of the anti-cryptococcal melanization response in *G. mellonella* (Fig. 3a, b). Using this technique, we visualized melanized nodules in situ that formed only during infection with *C. neoformans* and not in uninfected controls (Fig. 3c, d). These in situ melanized nodules (Fig. 3d–f) appeared very similar in size and shape to those that are collected from extracted hemolymph, which represented an ex vivo method of visualizing the nodules unique to fungal infection (Figs. 1a, 3g). The visual similarities between Fig. 3e, f and Fig. 3g clarified that the nodules observed in extracted hemolymph are generally representative of the entirety of nodules in the organism. Both the ex vivo and in situ techniques could be quantified to determine the average melanized nodule area and degree of melanization (Fig. 3h). However, the cleared tissue had some opacities or normally darker tissues (i.e. digestive tract contents, legs, prolegs, spiracles, cuticle pigmentation, etc.), which can result in the detection of dark particles even in the uninfected controls, albeit at a much lower frequency (Fig. 3h). Further, while there are some large *C. neoformans* nodules in situ that appeared aggregated (Fig. 3i, arrows), there was no clear anatomical tropism for nodule formation and the nodules are found throughout the larvae, implying that the infection was disseminated throughout the body of larvae, possibly through the insect's open circulatory system. The large, aggregated nodules were imaged along the Z-axis, which allowed for 3D reconstruction of the nodule for a better understanding of the native nodule structure compared to the ex vivo preparations compressed under a slide (Supplementary Movie 7). However, compared to the ex vivo experiments, the resolution of the melanin-encapsulated *C. neoformans* in situ

is limited, and variations in opacity and tissue thickness could interfere with measurements.

**Candida spp. interactions with the melanin-based immune response.** We employed the in vitro, ex vivo, and in situ techniques described above to gain insight into the host-microbe interactions of *C. albicans* with the *G. mellonella* melanin-based immune response. Using the ex vivo technique to analyze melanized nodules in infected hemolymph following extraction, we observed melanin-encapsulated *C. albicans* cells within nodule structures (Fig. 4a). These melanized nodules are like those that were observed during *C. neoformans* infection; however, the borders of the melanin itself appeared less distinct, blurry, and smudged. An additional difference from cryptococcal infection was the presence of filamentous *C. albicans* structures within the nodules. These hyphae or pseudohyphae were melanin-encapsulated, but seemingly to a lesser extent than the yeast morphology.

Analysis of the tissue of larvae infected with *C. albicans* for 24 h using the in situ tissue clarification technique (Fig. 4b) revealed groupings of the melanized nodules, often in long string-like patterns (Fig. 4b), that did not appear particularly associated with any organs or tissues (Fig. 4c, d, Supplementary Fig. 2a–f). Under higher magnification, we observed lightly pigmented hyphae and dark spherical melanized particles within the cleared larvae after 24 h of infection (Fig. 4c–f). These images validate that filamentation occurred within *G. mellonella*, which was previously observed with histology. The melanin-encapsulated fungi form large aggregates (Fig. 4c, d), which we initially thought could be indicative of a tropism for a specific tissue such as the chitinous trachea. However, upon dissection of uncleared infected larvae,

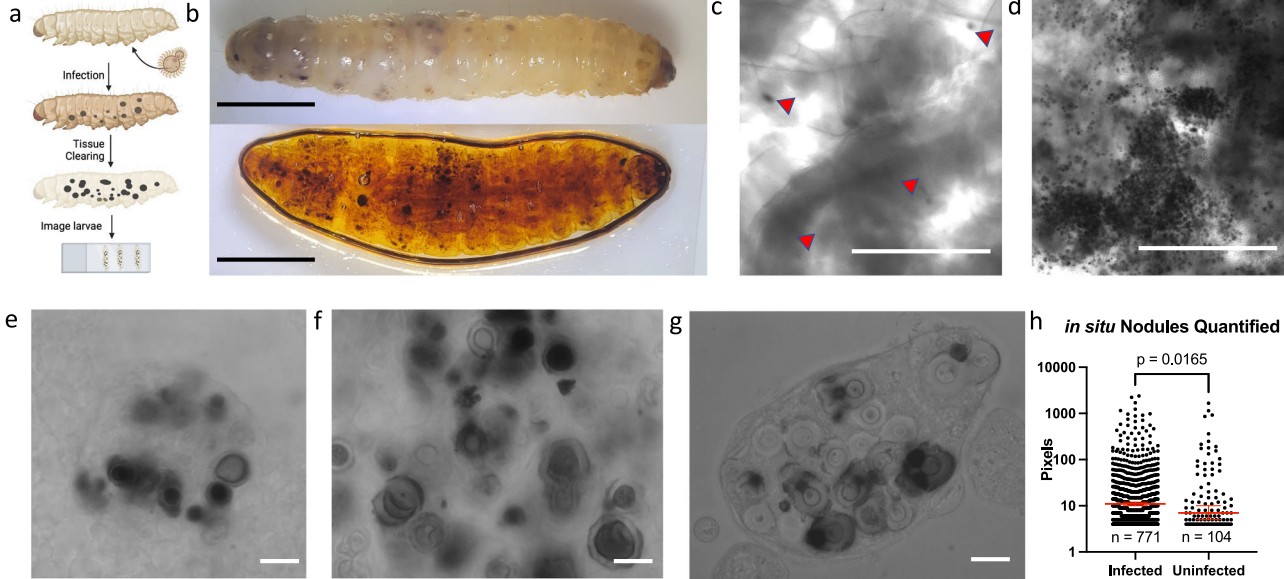

**Fig. 3 Using tissue clearing to visualize the melanin-based immune response against *C. neoformans* in situ.** Using tissue-clearing techniques (**a**), we allowed for better visualization of melanized nodules within the intact G. mellonella, as seen in a before (top) and after (bottom) view of the same infected larvae (**b**). Under microscopy, the uninfected larvae had little to no melanized nodules (red arrows indicate dark, irregular, or opaque areas in the tissue) (**c**), whereas the C. neoformans larvae had very clear and distinct nodules (**d**). These nodules were viewed under high magnification, where the nodule structure and encapsulated fungus were apparent (**e**, **f**). These structures were very similar in appearance to the nodules extracted from hemolymph (**g**). **h** The size of the melanized nodules were quantified using particle measuring software. Infected particles represent $n = 771$ independent particles, uninfected particles represent $n = 104$ independent particles, (Mann–Whitney test, $p = 0.016$). Error bars signify median with 95% Confidence Interval. **i** The melanized nodules of C. neoformans appeared to have no tissue tropism, but were found in distinct clumps or areas throughout the larvae (red arrows). **b**–**g**, **i** are representative images from three biological replicates. Scale bars in **e**–**g**, **i** represent 10 µm, **c**, **d** represent 500 µm, and **b** represent 50 mm.

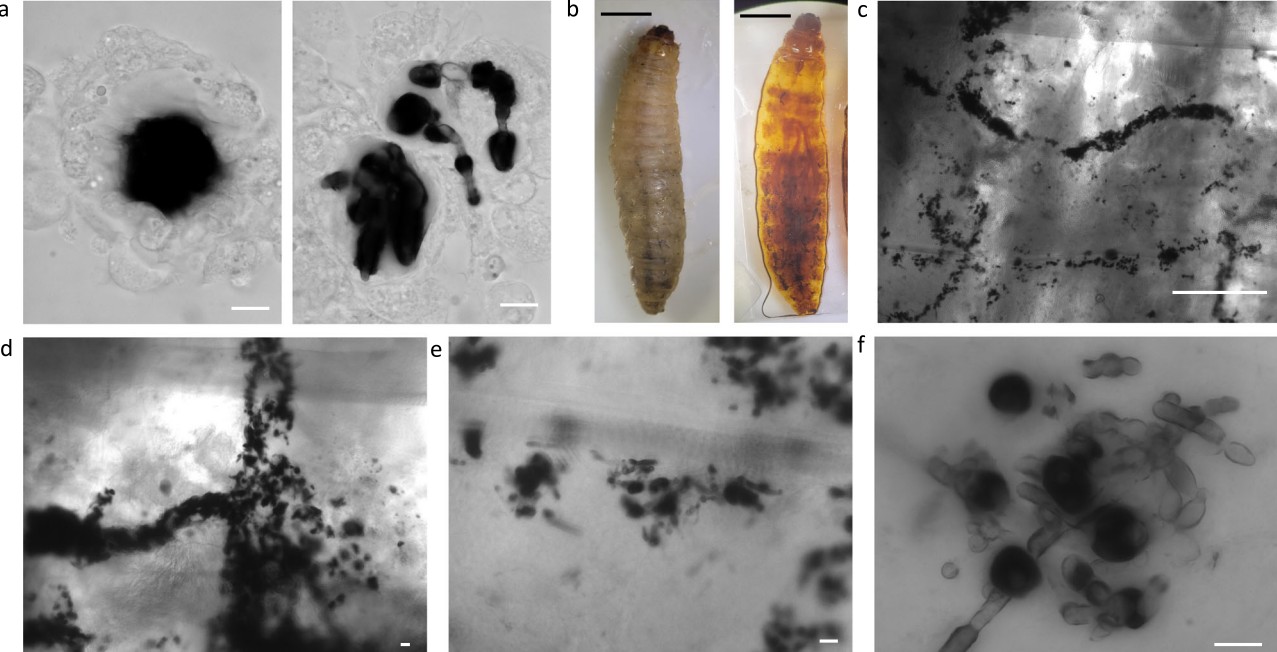

**Fig. 4 Visualizing the melanization response to infection with *Candida albicans*. a** Similar to the hemolymph from larvae infected with C. neoformans, we are able to see melanized nodules within the hemolymph. The melanized spot within these nodules appears more diffuse/less defined, and in some, the presence of less-melanized hyphae is distinct (red arrows). **b** We can also use tissue clearing to visualize the melanized nodules during C. albicans infection. **c**, **d** The melanized C. albicans seem to cluster in specific areas, in long strips within the larvae (yellow arrows). **e**, **f** Under higher magnification, we can see hyphal structures of C. albicans, which corresponds to what is previously known about C. albicans morphology in G. mellonella. Interestingly, the hyphae appear less melanized (red arrows) compared to the spherical yeast (white arrows) (**f**). All panels show representative images from 3 biological replicates. Scale bars in **a**, **d**–**f** represent 10 µm, in **c** represent 500 µm, and in **b** represent 50 mm.

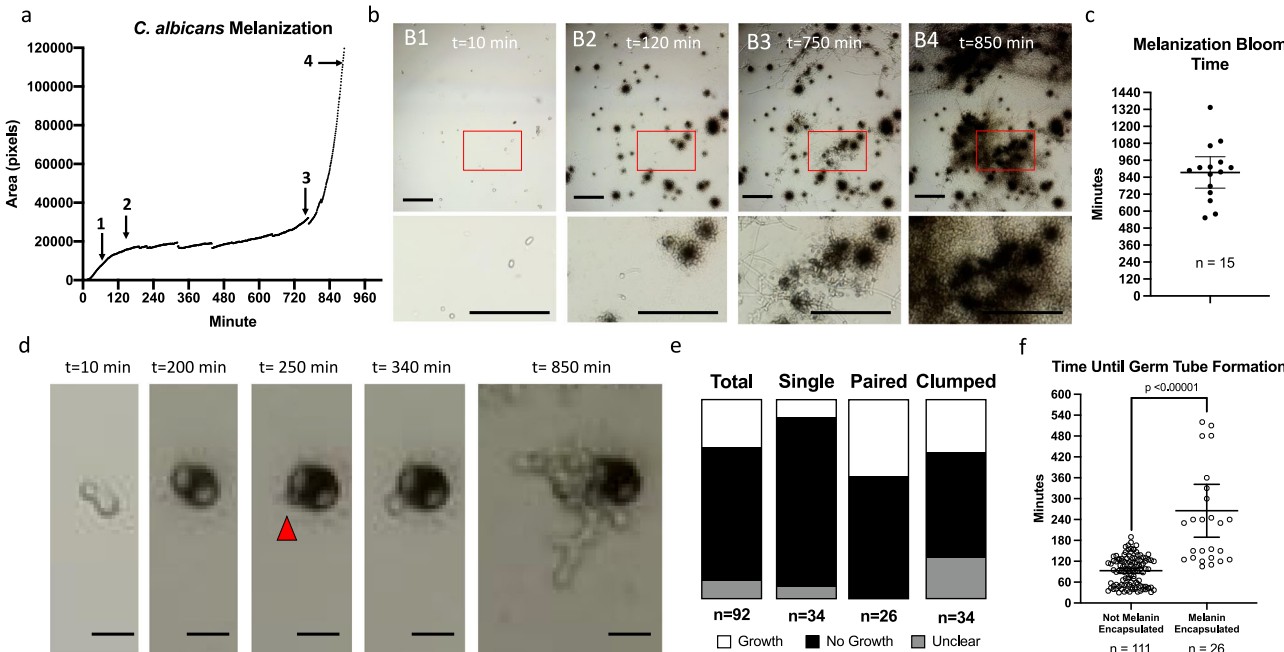

**Fig. 5 Using time-lapse microscopy to gain insight into *C. albicans* yeast and hyphae interactions with the melanization immune response. a** Kinetics of the melanization reaction within hemolymph incubated with C. albicans yeast, with a distinct tri-phastic response in which the yeast starts off non-encapsulated with melanin (1), followed by an initial peak of the melanization reaction (2), an extended plateau (3), and a greater second peak of the melanization reaction (4). **b** Using microscopy, we saw that the yeast start off non-melanized (B1), and most reached their peak melanin-encapsulation by 120 min (B2). After which, the yeast began to filament, and hyphae grew between 120 and 750 min with minimal melanin encapsulation occurring. At 750 min (B3), blastoconidium (yeast) began to form on the hyphae, which then corresponded to a rapid increase in melanization (B4). Scale bars in **b** indicate 100 μm. **c** The time until this melanin bloom was ~14 h ($n = 15$ biologically independent movies, mean = 874 min, 95% Confidence Interval 762–985 min). **d** Through these movies, we also saw that some melanin-encapsulated yeast were able to survive the immune response and grew from underneath the layer of pigment. The melanin-encapsulated C. albicans in panel **d** began to grow through the melanin by 250 min. Scale bars in **d** indicate 10 μm. **e** Overall, ~23% of melanin-encapsulated yeast were able to grow following melanin-encapsulation. Pairs of melanin-encapsulated yeast had the highest percentage of growth following melanin encapsulation, with a 33% frequency. **f** In the yeast that grew following melanin-encapsulation, the time until germ tube formation was delayed 265 min on average ($n = 26$ independent melanin-encapsulated fungi), while in non-encapsulated yeast, the average time was 92 min ($n = 111$ independent non-encapsulated fungi) (Two-tailed unpaired *t*-test, $p < 0.00001$, $t = 8.878$, $df = 135$) and error bars indicate mean with 95% Confidence Interval. **a**, **b**, **d** show representative data from three biological replicates. Each data point in **c** represents an independent biological replicate. **e**, **f** show data from 4 biological replicates.

there did not appear to be an association of these clusters with any specific tissues (Supplementary Fig. 2a–f). Differences in pigmentation between the two *C. albicans* morphologies, particularly as seen in the Z-projection in (Fig. 4f), indicated that the hyphae were encapsulated with less melanin during infection compared with the yeast form of the fungus. One potential bias in interpreting this data is that since we are only looking at melanin pigmentation, we are likely missing any non-melanin-encapsulated fungi which would be obscured within the insect tissue.

We used the in vitro time-lapse microscopy to observe the melanization dynamics of *C. albicans* in hemolymph. As seen earlier (Fig. 2h), *C. albicans* yeast triggered a more robust melanization response than *C. neoformans* (Fig. 5a, 5b1). In time-lapse microscopy performed without the addition of insect hemocytes, we observed that the *C. albicans* began to grow in filamentous forms (Fig. 5b2), consistent with the importance of filamentation in the pathogenesis of *C. albicans* within *G. mellonella* and the in situ data (Fig. 3)[23]. In mammalian hosts, *C. albicans* filamentation is previously shown to be triggered by serum, neutral pH, and temperature[41,42]. However, in the *G. mellonella* system, filamentation in vitro does not occur when the *C. albicans* is only incubated with hemocytes without hemolymph, indicating that a component of the hemolymph is necessary for the morphological switch. Interestingly, as the

time-lapse movie progressed, we observed that the hyphae did not get encapsulated by melanin in comparison to the yeast form of *C. albicans* (Fig. 5b2). After about 12 h of filamentous growth, we observed the formation of blastoconidium (yeast) along the hyphae (Fig. 5b3). The formation of these yeast cells then corresponded with a subsequent bloom of melanization (Fig. 5A, 5B4, Supplementary Movie 8). A similar temporal progression of *C. albicans* morphology and melanin-encapsulation is seen in *C. albicans* infected larvae dissected at various timepoints post-infection (Supplementary Fig. 2g). The average time of this melanin bloom was about 840 min, with a 95% confidence interval between ~720 and 960 min (Fig. 5c).

We observed that some of the melanin-encapsulated *C. albicans* yeast survived the immune reaction and then underwent hyphal and/or pseudohyphal growth (Fig. 5d, Supplementary Movie 9). This occurred in about 23% of melanin-encapsulated yeast, with 8% of single yeast and 38% of budded/pairs of yeast being able to escape (Fig. 5e) The time until hyphal or pseudohyphal growth was significantly delayed in the melanin-encapsulated cells; the average time for a non-melanin-encapsulated *C. albicans* cells to begin filamentation is 92 min, while the melanin-encapsulated counterparts take 265 min, with some taking as long as 520 min (Fig. 5f). This delay could be reflective of physical barriers as the fungus breaks through the melanin layer and/or delays in initiating cellular growth because of cell damage caused by the immune response.

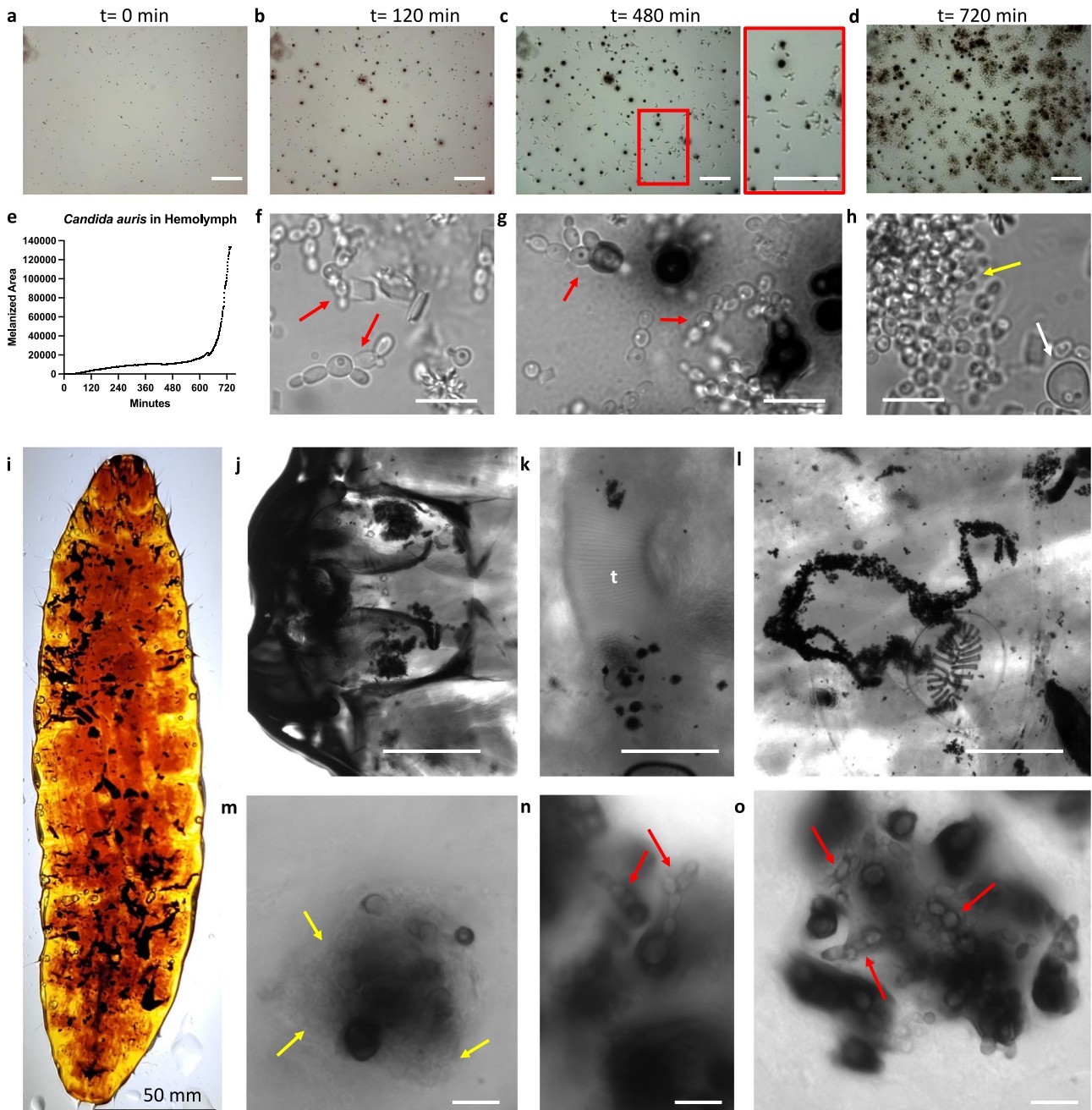

**Fig. 6 *Candida auris* produces pseudohypae in response to hemolymph.** Following incubation of *C. auris* with hemolymph, the yeast get encapsulated with melanin by 120 min (**a**, **b**). Pseudohyphae form clearly by 480 min, while not eliciting a melanization response (**c**, inset); however, after ~720 min, there is a sharp increase in melanization within the hemolymph (**d**). A representative graph of the melanization in response to *C. auris* over time is shown in **e**. The cell morphologies were confirmed as pseudohyphae through light microscopy at ×100 magnification (e.g., red arrows), while the presence of enlarged giant cells was occasionally noted (**h**, white arrows) and dense clusters of yeast (**h**, yellow arrows). **i** Melanized aggregates of *C. auris* can be seen in *G. mellonella* larvae following tissue clearing. Melanized clusters of fungi can be seen in the head capsule (**j**), associated with trachea (**k**, t label) and freely within the tissue (**l**). In these melanized areas, we can see aggregates of yeast (**m**, yellow arrows) and pseudohyphae (**n**, **o**, red arrows). All panels are representative images and data from three independent replicates. Scale bars in panel **i** represent 50 mm, **j**–**l** represent 500 µm, **a**–**d** represent 100 µm, while **f**–**h**, **m**–**o** represent 10 µm.

Similar to our findings with *C. albicans*, we observed that *Candida auris* yeast activated melanization of hemolymph (Fig. 6a, b). This was followed by formation of melanin-evasive pseudohyphae (Fig. 6c) and a bloom of melanization (Fig. 6d–e, Supplementary Movie 10). Formation of pseudohyphae were confirmed visually by conventional light microscopy (Fig. 6f–g). Enlarged giant cells—cells that are larger than the typical *C. auris* yeast—measuring roughly 8 µm were observed

sporadically (Fig. 6h). Clearing of *C. auris* infected *G. mellonella* revealed large aggregates of melanin-encapsulated fungi (Fig. 6i–l), including localization within the head capsule (Fig. 6j), adjacent to trachea (Fig. 6k), and not associated with noticeable tissue structures (Fig. 6l). Similar to what was seen in vitro, in situ we saw large aggregates of non-melanized yeast (Fig. 6m, yellow arrows) and pseudohyphal fungal morphology (Fig. 6n, o).

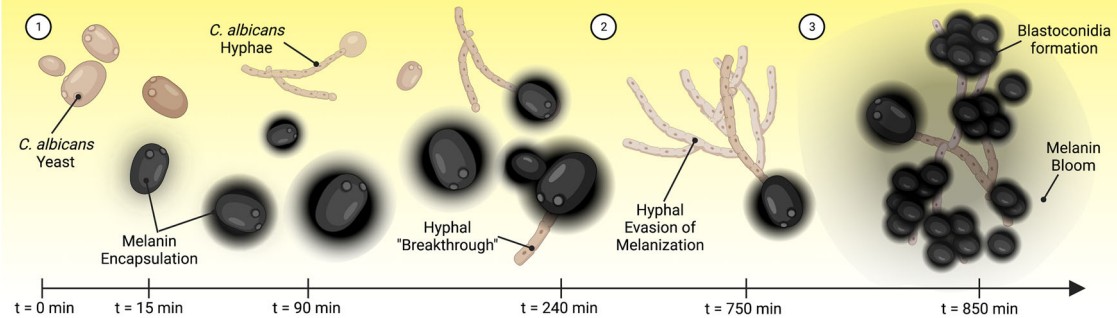

**Fig. 7 Overview of the growth of *C. albicans* within *G. mellonella* and the fungus' interaction with the melanization immune response.** (1) Within 15 min of exposure to hemolymph, immune melanin forms around the C. albicans yeast. Following melanin encapsulation, ~23% of yeast form a germ tube that grows and escapes the melanin encapsulation. Non-melanin-encapsulated yeast form germ tubes faster and more frequently. (2) Hyphal growth continues with minimal melanin encapsulation occurring on hyphae. (3) Yeast (blastoconidia) begin to form on hyphae after 12–16 h, which triggers a large-scale bloom in the melanization response.

Altogether, these data demonstrate three phases of the *C. albicans*-immune melanization interactions: (1) yeast become encapsulated with melanin, with nearly 25% surviving and breaking through the pigment, (2) cells undergo a yeast-to-hyphal transition, with the hyphal and pseudohyphal cells evading the melanization immune response; and (3) filamentous *C. albicans* begins to produce more yeast cells (referred to as blastoconidia or blastospores), which then causes a second bloom of melanization to occur (Summarized in Fig. 7). While host melanization and fungal filamentation have been well-reported during the course of *C. albicans* infection[23,43–46], we report for the first time, to the best of our knowledge, that the hyphae and pseudohyphae are melanin-evasive. Although the presence of blastoconidia has been reported in *G. mellonella* infected with *C. albicans*[43], these data indicate that lateral blastoconidia growing from hyphae induce a strong second wave melanization response.

## Discussion

Melanin has been appreciated as a key part of the insect immune defense against microbes and parasites for the greater part of the past century[13,47]. Immune melanization has been implicated as a major process in neutralizing entomopathogenic fungi upon infection[48]. The insect phenoloxidase and the melanization cascade produce toxic intermediates such as dihydroxyindole and high levels of oxidative stress that can overwhelm and kill the fungus or microbe in vitro[13,17]. However, there have been no in vivo studies showing that melanin encapsulation within nodules directly kills fungi during these immune reactions within the insect. Here, we fill that gap by showing that melanization within nodules is associated with the death of *C. neoformans* using a GFP viability reporter assay and provide additional in vitro data for a fungicidal role in immune defense.

During cryptococcal infection of *G. mellonella*, melanin encapsulation of the fungus within nodules was associated with diminished or lost fluorescence signal in these GFP-expressing *C. neoformans* strains. Additionally, the melanin-encapsulated fungi that remained GFP-positive had weaker signals and the intensity of the GFP signal was more intense for the non-melanin-encapsulated fungi within the nodules. The expression of GFP in these cells is under the control of an actin promotor, and while actin is generally presumed to be constitutively expressed in cells, growth conditions have been shown to lead to some alterations in cryptococcal actin expression[49,50]. If the environmental conditions within the nodule abolished actin expression in some cells without killing the fungus, we would expect that condition to equally affect the melanin and non-melanin-encapsulated fungi,

and as a result, see similar GFP-negative: GFP-positive ratios between the melanin-encapsulated and not melanin-encapsulated cells. The association between melanin encapsulation and disappearance in GFP fluorescence provides strong evidence for the notion that the melanization reaction kills fungal cells during infection. This is the first evidence, to the best of our understanding, that *G. mellonella* immune melanization directly and effectively neutralizes *C. neoformans* during infection and demonstrated that melanin encapsulation results in fungal death within the insect. Previously, the death of microbes, specifically bacteria, was attributed to the enzymatic activity of the melanin-producing phenoloxidase (PO) in an in vitro reaction[16]. In addition to our association of melanin encapsulation and fungal death in vivo, we sought to reproduce these results in vitro using extracted hemolymph in buffer. We used the PO-specific inhibitor, phenylthiourea (PTU), we found that PO-inhibited wells of hemolymph had higher recoverable CFUs of *C. neoformans* compared to the uninhibited wells. The inverse correlation of melanization with CFUs further supports the claim that melanin plays a role in neutralizing *C. neoformans*. Since we only assayed CFUs from these in vitro experiments, we cannot determine whether the melanization in the in vitro experiments directly killed the fungus or just inhibited fungal growth.

When attempting to replicate our results with a known cell viability dye, PI, we were unable to obtain staining of fungal cells within nodules. The absence of PI staining for fungal cells in nodules suggests that PI was unable to reach the center of the nodules where the fungi are found and shows that some of the hemocytes involved in surrounding the fungus may undergo cell death in the process. The permeability or access issues that may arise when using added dyes to measure microbial viability within the nodule show the usefulness of using a live-dead indicator that is endogenously present within the fungus, such as the constitutively expressed GFP.

In addition to studying the extracted hemolymph, we used a developed in vitro time-lapse microscopy assay, which allows observation of the melanization response in real time, while also using the same population of hemocytes and stock of hemolymph to reduce inter-larval variation in the melanization response. Since we only used hemocytes that adhere to the coverslip, we are making an assumption that the hemocyte population is representative, while there might be a predilection for the adhesion of some hemocyte subtypes compared to others. We investigated the impact of fungal melanins on the insect immune response. We found that isolated fungal melanins, termed melanin ghosts, activated the melanin-based immune response whereas the heat-

killed *C. neoformans* did not. This suggests that fungal melanins can activate the immune system which could help promote fungal clearance. This is interesting in the context of naturally occurring fungal pathogens of insects, which tend to have a white color and/ or do not naturally produce melanin pigment, such as *Beauveria bassiana* and *Metarhizium anisophilae*[24,25]. While these fungi have undergone many adaptations to effectively infect insect hosts[51], their lack of melanin may be due to an additional evolutionary pressure that selects for fungi that produce less fungal melanin, and thus elicit a less robust melanin-based immune response. Since melanin is a component of the insect wound response, it is possible that these exogenous melanins are recognized by the insect as a damage-associated molecular pattern and launch an inappropriate wound repair response.

We found that the *lac1Δ* mutant, which is unable to produce the melanin-producing enzyme laccase, causes less melanization in the hemolymph. This implies that some of the melanization that occurred was due to fungal laccase-catalyzed melanization using host-derived catecholamines in the hemolymph. Another possible mechanism is that laccase produces fungal melanins, which in turn activates the melanization response of insects as we had seen with the isolated fungal melanins. In the *lac1Δ*, this fungal melanin trigger would be absent and thus induce a less robust melanization response. These findings are also consistent with the observation that during *B. bassiana* infection of *G. mellonella*, laccases play a role in virulence by oxidizing the hemolymph catecholamines and preventing them from producing antifungal melanization and reducing the oxidative burden on the fungus[24]. It is also worth noting that the *lac1Δ* mutant is less virulent in *G. mellonella* infections compared to the parental strain[19]. Together, these observations paint a nuanced picture of the role that laccase and fungal melanin play during fungal pathogenesis in *G. mellonella*—both fungal melanin and fungal laccases activate the melanin-based immune response, while fungal melanins are associated with decreased virulence, fungal laccases enhance virulence. We note that laccase is secreted by *C. neoformans* and is found in extracellular vesicles, which could transport laccase away from the fungal cell and reduce the antifungal damage from its effects on triggering insect immune melanization. We compared the amount of melanization that *C. neoformans* triggers with the amount triggered by other fungal species such as *C. albicans*. The differences in hemolymph-induced melanization during exposure to *C. albicans* and *C. neoformans* were previously described[39], and our results confirm those findings. The smaller magnitude of the melanization reaction in response to *C. neoformans* may be due to cell wall pathogen-associated molecular patterns being hidden by the fungus' polysaccharide capsule, which is known to help the fungus evade host immunity in insects and mammals[3,52].

The second method used to evaluate the melanization response to *C. neoformans* was tissue clearing, which enabled us to visualize melanized nodules in situ deep within the larvae. We modified a recently developed protocol[40] to view the nodules that formed during infections, and saw the native structures of the nodules and their anatomical location in the larvae. This offers an advantage over dissection of uncleared larvae, because during the dissection process: (1) the tissue organization is disrupted, (2) some organs such as the nerve cord and cardiac system might be disrupted, and (3) the geography of infection patterns may not be apparent. Additionally, melanized nodules may not be visible within or behind opaque tissues and organs. Tissue dissection of opaque larvae was helpful when evaluating tissue tropism since tissue boundaries may not be fully visible in clarified larvae. A bias involved in studying fungal infections using both tissue clearing and dissection is that the non-melanized nodules or fungi may be missed, as unpigmented fungi will likely blend in with

surrounding tissue. However, in the clearing method, we viewed the nodules throughout the entire depth of the larvae at a low to moderate (×4–40) magnification using light microscopy. However, the objective and microscope limitations only permitted imaging the superficial melanized nodules at ×100 magnification, which provided a lower resolution of the nodules compared to the imaging of the extracted hemolymph. While in the case of *C. neoformans*, the nodules within the hemolymph appeared congruent to those viewed in situ, that might not always be the case. Nodules in extracted hemolymph during other fungal infection may not be entirely representative of those found throughout the entire larvae, so only viewing the hemolymph nodules may give a biased understanding of the fungal infection.

We also examined the melanization response to *Candida albicans* infection. *C. albicans* is known to trigger large-scale systemic melanization in *G. mellonella* larvae[39,53]. Like *C. neoformans*, we found melanized nodules in the hemolymph from larvae infected with *C. albicans*. Interestingly, the center of these nodules had melanized and smoothened areas that seemed more amorphous than those seen with *C. neoformans*, and additionally, we saw hyphal structures appeared less melanized than the spherical yeast-like structures. Using the tissue clarification method, we noted that the melanin-encapsulated *C. albicans* formed large rope-like aggregates without tissue tropism, with yeast being preferentially melanized over hyphal cells. Using in vitro time-lapse microscopy, we found that rapid melanization occurred, even in the absence of hemocytes. Additionally, after the melanization plateaus, the surviving fungus can break free from the melanin encapsulation and undergo melanin-evasive filamentation. This is followed by production of laterally budding blastoconidium and a bloom in melanization around these newly formed yeast cells. Similar fungal morphologies and timelines were observed in dissected infected larvae, although the temporal kinetics were less resolved and identification of blastoconidium was less clear. Together, these data paint an interesting picture and allow insight into the pathogenesis of *C. albicans* within *G. mellonella* host. Hence, it appears that the melanin encapsulation can clear most of the yeast upon infection; however, cells that survive can then filament and evade subsequent melanin-mediated killing. The hyphae are known to penetrate and infect organs within the insect[23]. The hyphae then produce yeast, which again triggers a burst of melanization that would likely cause damage to the surrounding tissue and eventually death of the organism.

Notably, we observed the formation of melanin-evasive pseudohyphae in *C. auris* when exposed to the *G. mellonella* hemolymph. These findings corroborate previous studies showing that *C. auris* forms pseudohyphae during *G. mellonella* infection[54], and indicate that the stress and cues from the hemolymph can induce morphological changes in multiple fungal species. Pseudohyphae and enlarged giant cells in *C. auris* have been previously linked to exposure to genotoxic stress, indicating that the melanization reaction can induce DNA damage[55]. Following tissue clearing of *C. auris* infected larvae, we observed pseudohyphal structures in situ. We also observed melanin-encapsulated fungi within aggregates surrounding trachea, as previously reported[54,56], although the frequency of this association was difficult to ascertain, with aggregates of melanin-encapsulated *C. auris* was also found unassociated with trachea, including in the insect's head capsule.

In summary, we found evidence that the *G. mellonella* wax moth directly kills *C. neoformans* by encapsulating it with melanin in vivo using a GFP-expressing strains where fluorescence indicates viability. This association between melanin encapsulation and reduced viability provides direct evidence for fungal killing via melanin encapsulation in vivo. We also describe three

different methodological approaches for studying the melanization response to fungi in *G. mellonella* and employ these techniques to study *C. neoformans*, *C. albicans*, and *C. auris'* interactions with the melanin-based immune response. With *C. neoformans*, we show that both fungal melanins and fungal laccases can activate the insect's melanization immune response, furthering our understanding of how these fungal components interact with insect immunity and alter the fungus' pathogenesis. In *C. albicans*, we were able to observe how some melanin-encapsulated yeasts were able to breakthrough the melanization, and form melanin-evasive hyphae and pseudohyphae during infection. The direct association of insect melanization with antifungal defense further heighten concern that pesticides that inhibit the melanin reaction[30] could have untoward and unpredictable effects on insect populations.

## Methods

**Biological materials.** *G. mellonella* last-instar larvae were obtained through Vanderhorst Wholesale, St. Marys, Ohio, USA. For the tissue-clearing experiments with *Candida auris*, *G. mellonella* larvae were obtained from BestBait.com, Marblehead, OH, USA. *C. neoformans* strain H99 (serotype A), *C. neoformans* acapsular mutant *cap67Δ*, *C. neoformans* strain H99-GFP[57], *C. neoformans* lac1Δ mutant (from the 2007 mutant library created by Dr. Jennifer K Lodge, and was obtained from the Fungal Genomics Stock Center), *Candida albicans* strain 90028, and *Candida auris* CDC 388 (B11098) were kept frozen in 20% glycerol stocks and subcultured into yeast peptone dextrose (YPD) broth for 48 h at 30 °C prior to each experiment. For H99-GFP infections, frozen stock was streaked out first onto YPD agar, and green colonies were inoculated into YPD broth for 48 h at 30 °C prior to each experiment. The yeast cells were washed twice with PBS, counted using a hemocytometer (Corning, New York, USA), and adjusted to $10^7$ cells/ml for an injection inoculum of $1 \times 10^5$ cells/larva. *C. albicans* infections were performed at $5 \times 10^5$ cells/larva with the inoculum suspension being diluted to $5 \times 10^7$/ml.

**Extraction of hemolymph from fungal-infected *Galleria mellonella* larvae.** Infection of *G. mellonella* larvae was performed as previously described[30]. In summary, washed *C. neoformans* were resuspended to $10^7$ cells/ml, and 10 μl were injected in the right rear proleg of larvae ranging from 175 to 225 mg, for an infectious inoculum of ~$10^5$ cells/larva. *C. albicans* was washed and resuspended in $5 \times 10^7$ cells/ml, and larvae were injected with 10 μl of culture for an infectious inoculum of ~$5 \times 10^5$ cells/larva. Infected larvae were then incubated at 30 °C. 24 h following infection, or 72 h in the case of noted time course experiments with *C. neoformans*, larvae were removed from incubator, and hemolymph was extracted by puncturing the right rear proleg and/or side of the larvae with an 18 G needle. Removed hemolymph from three larvae was collected directly into 1 ml anticoagulation buffer at room temperature (98 mM NaOH, 186 mM NaCl, 1.7 mM EDTA, and 41 mM citric acid, pH 4.5)[58]. Hemolymph was centrifuged for 5 min at $4000 \times g$ and resuspended in 200 μl insect physiological saline (IPS) (150 mM sodium chloride, 5 mM potassium chloride, 7.21 mM calcium chloride, 1 mM sodium bicarbonate, pH 6.90—adapted from refs. [59–61]). Samples were placed on slides and nodules were imaged using Olympus AX70 microscope with a ×100 oil immersion objective.

**C. neoformans GFP viability assay.** H99-GFP strain was streaked from frozen stock on YPD agar and incubated at 30 °C. 2 ml YPD was inoculated with H99-GFP and incubated for ~18 h at 30 °C with rotation. Culture was diluted to OD 0.5 and 100 μL was incubated at 70 °C for 1 h using a thermocycler. One hundred microliters of untreated and heated samples were stained with 10 μg/ml propidium iodide (Invitrogen). Ten microliters of stained samples were loaded onto a hemocytometer and imaged using a ×10 objective and Zeiss AxioImager M2 (×60 Olympus objective) equipped with a Hamamatsu Orca R2 camera and Volocity Software (Perkin Elmer). Images were analyzed using ImageJ/FIJI software. Fluorescence channel images were processed by adjusting the minimum pixel value to 10 and maximum to 90. Number of fluorescent cells for each channel were counted using Measure Particles. Number of double fluorescence positive and double fluorescence negative cells were enumerated manually.

**C. neoformans GFP fungal survival assay in vivo.** Washed H99-GFP strains of *C. neoformans* were resuspended to $10^7$ cells/ml, and 10 μl was injected in the right rear proleg of larvae ranging from 175 to 225 mg, for an infectious inoculum of ~$10^5$ cells/larva. Infected larvae were then incubated at room temperature 30 °C for 24–72 h as noted. Hemolymph was extracted by puncturing the larvae with an 18 G needle in their right rear proleg, and hemolymph was collected into anticoagulation buffer, centrifuged for 5 min at $4000 \times g$, and resuspended in 200 μl insect physiological saline. Melanized nodules were visualized using an Olympus AX70 microscope with 488 excitation/520 nm emission fluorescence microscopy to visualize the GFP signal. Images were taken at ×100 magnification at the same

exposure, and manually marked as positive or negative for GFP fluorescence, and melanin-encapsulated or unencapsulated. For fluorescence and melanin intensity measurements, images were analyzed using the Measure tool in FIJI[62], and the 8-bit mean gray value of each cell was measured in both channels. The region selected for the melanin measurements extended the edge of the fungal capsule, and the GFP intensity measurements were from selections limited to the fluorescent cell's body. This protocol is summarized in Supplementary Fig. 3a.

**Phenoloxidase inhibition and fungal survival assay in vitro.** Serial dilutions of phenylthiourea (PTU) were performed in 100 μl IPS buffer, to which 5 μl of $10^6$ cells of *C. neoformans* hemolymph was added. *G. mellonella* hemolymph was extracted by puncturing the larvae with an 18 G needle in their right rear proleg, and letting the hemolymph drip into insect physiological saline. The mixture was mixed by pipetting up and down and 100 μl was added to each well. The mixture was incubated at room temperature for 24 h protected from light. Following the incubation, the contents of each well were resuspended and diluted 1:16 in PBS. From the dilution, 5 μl was spotted on a Sabouroud agar plate. The plate was incubated at 30 °C for 24 h and colonies were enumerated under a dissection microscope.

**Tissue clearing of *Galleria mellonella* following fungal infection.** *C. neoformans* was washed in PBS, counted, and resuspended to $10^7$ cells/ml, and 10 μl was injected in the right rear proleg of *G. mellonella* larvae ranging from 175 to 225 mg, for an infectious inoculum of ~$10^5$ cells/larva. *C. albicans* was washed and resuspended in $5 \times 10^7$ cells/ml and larvae were injected with 10 μl of culture for an infectious inoculum of $5 \times 10^5$ cells/larva, while *C. auris* was resuspended in $10^8$ cells/ml and larvae were injected with 10 μl of culture for an infectious inoculum of $10^6$ cells/larva. Following infection (5 days for *C. neoformans*, 24 h for *C. albicans*, and 48 h for *C. auris*), groups of three larvae were removed from incubator and injected with 10 μl of 1 M ascorbic acid to inhibit new melanization and oxidation of endogenous catecholamines during the tissue-clearing process. Ten minutes following the ascorbic acid injection, larvae were placed at −20 °C for 15 min to euthanize them, then injected with an additional 10 μl of 1 M ascorbic acid. Larvae were immediately placed in 40 mL of 4% paraformaldehyde to fix overnight at 4 C. Fixed larvae were washed in PBS, and serially dehydrated in 50, 75, 100, and 100% methanol for 1 h each at 4 °C. Larvae were then incubated for 1 h in 50% methanol and 50% BABB solution (2:1 Benzyl benzoate: Benzyl alcohol), and cleared in 100% BABB solution as previously described[40]. Following 5–7 days of tissue clearing, larvae were removed from the BABB solution and pressed between two glass microscope slides. Once flattened, a coverslip was placed on top of the larvae and parafilmed into place. Larvae were imaged using Olympus AX70 microscope with ×4, ×20, and ×100 objectives.

**Imaging Galleria mellonella hemocytes in vitro.** To collect and isolate hemocytes *G. mellonella* larvae were surface sterilized in two sequential baths of 70% ethanol, followed by 10% bleach, then dried on sterile paper towels. Five to ten drops of hemolymph were extracted as described above into room temperature anticoagulation buffer and inverted three times. Hemolymph was centrifuged at $400 \times g$ for 4 min, the supernatant was removed, and the hemocytes were resuspended in 1 ml anticoagulation buffer and centrifuged again at $400 \times g$ for 4 min. The supernatant was completely removed and hemocytes were resuspended in 200 μl of insect physiological saline (IPS). The 200 μl suspension of hemocytes were added to the coverslip of a 14 mm diameter microwell MatTek dish (Catalogue #P35G-1.5-14-C, MatTek) and allowed to settle for 10 min Following the 10 min, the buffer and unsettled hemocytes were removed, and the coverslip was washed four times with 1 ml of IPS. The hemocytes were seeded into the coverslip at a cell density of $1.5 \times 10^6$ cells/ml and the resulting hemocyte density after washing was ~$2–3 \times 10^3$ cells/mm$^2$.

While the hemocytes were being isolated, cell-free hemolymph was being prepared. Approximately 10 drops of hemolymph were removed from *G. mellonella* larvae and collected directly into 1 ml IPS. To remove hemocytes, the mixture was filtered using a 0.22 μm syringe-driven PVDF filter. Cell-free hemolymph was stored up to a week at -80C. Penicillin-Streptomycin (Gibco, Thermo Fisher) antibiotic was added at 1× concentration to the cell-free hemolymph. For experiments looking at the interaction of hemocytes with fungi or a virulence factor, the cells or component were added at this stage.

Following the hemocyte washes, 1 ml of cell-free hemolymph was added to the entirety of the MatTek dish, followed by an addition 1 ml of IPS. The MatTek dish was covered and imaged using the OpenFlexure microscope and software and time-lapse microscopy was performed every minute for 16–24 h[63]. Control experiments without added fungus or fungal components, or with anticoagulation buffer to inhibit melanization and coagulation, were performed (Supplementary Movies 11 and 12). This protocol is summarized in Supplementary Fig. 3b.

All time-lapse data were analyzed using FIJI[62] and particle measurements were made by converting the image sequence to 8-bit, setting a threshold of 0–50 gray value, and analyzing any particle over the size of four pixels[2]. Measuring the time until germ tube formation was done manually by recording the frame in which the first of the germ tube was visible.

**Melanin ghost isolation**. *C. neoformans* cultures were grown in minimal media with 1 mM L-DOPA for 7 days at 30 °C. Cells were collected and mixed 1:1 with 12 N hydrochloric acid (HCl), for a final concentration of 6 N HCl. Cells were heated for 1 h at 85 C under constant shaking at 350 RPM. Control cells from the acapsular mutant grown for 5–7 days in minimal media at at 30 °C without L-DOPA were heat-killed for 1 h at 85 °C in PBS. Cells were washed twice in PBS and subsequently used in the time-lapse microscopy.

**Statistics and reproducibility**. Statistical tests were performed using GraphPad Prism Version 8. All error bars represent the mean with 95% Confidence Interval with the exceptions of the ratio comparison of melanization induced by H99 versus *lac1Δ* where the error bars represent the ±Standard Deviation from the mean, and the size comparison of in situ nodules where the error bars represent the 95% Confidence Interval of the median. To determine the effect of melanin-encapsulation on fungal viability within nodules, we performed a Chi-Squared analysis for each timepoint and temperature condition tested. The Chi-square table was set up with melanin-encapsulated and non-encapsulated on one side of the contingency table and GFP-positive and GFP-negative on the other. The ratio comparison of melanization induced by H99 versus *lac1Δ* was analyzed using a one sample t-test with comparisons to the theoretical value of 1, which would indicate no difference between the H99-induced melanization and *lac1Δ*-induced melanization. The size comparison of in situ nodules was performed using a two-tailed non-parametric Mann–Whitney Test comparing the particles from the uninfected versus infected groups. The comparison of the time it takes for the melanin-encapsulated versus non-encapsulated to escape melanization was done using an unpaired two-tailed *t*-test. The linear regression showing the relationship between CFUs and pigmentation of the hemolymph in vitro was performed using a simple linear regression, with the dotted lines representative of the 95% Confidence Intervals. Non-quantified data, such as microscopy images, were performed in replicates and representative images are shown, as noted in the figure legend.

**Reporting summary**. Further information on research design is available in the Nature Portfolio Reporting Summary linked to this article.

## Data availability

The datasets generated during and/or analyzed during the current study are available in the Figshare repository using the following link: https://figshare.com/projects/Galleria_mellonella_immune_melanization_is_fungicidal_during_infection/138123 [64].

## Code availability

The code used in this study is previously published open source code[63] and can be accessed using the following link: https://openflexure.org/software/raspbian-openflexure/.

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

## Acknowledgements

We would like to thank the entire Casadevall Lab for their contributions during lab meetings and other discussions of this project. We would like to thank Maryann Smith, Thomas Hitzelberger, and Kathy Spinnato for placing the years of weekly *G. mellonella* orders. Illustrations were created using Biorender.com. D.F.Q.S., Q.D., and A.C. are funded by National Institute of Allergy and Infection Disease R01 AI052733, R01 AI152078, and the National Heart, Lung, and Blood Institute R01 HL059842. D.F.Q.S. is funded by National Institutes of Health 5T32GM008752-18 and 1T32AI138953-01A1. J.M.H. and M.K. are funded by National Institutes of Health R56 AI168539 and R21 AI144373. The funders had no role in study design, data collection and analysis, decision to publish, or preparation of the manuscript.

## Author contributions

Conceptualization—D.F.Q.S., Q.D., M.K., J.M.H., and A.C.; Methodology—D.F.Q.S., A.C., Q.D., and M.K.; Software—Q.D.; Validation—D.F.Q.S.; Formal analysis—D.F.Q.S. and M.K.; Investigation—D.F.Q.S. and M.K.; Resources—A.C. and J.M.H.; Data curation —D.F.Q.S.; Writing (original draft)—D.F.Q.S. and A.C.; Writing (review and editing)— D.F.Q.S., Q.D., M.K., J.M.H., and A.C.; Visualization—D.F.Q.S. and M.K.; Supervision— D.F.Q.S., J.M.H., and A.C.; Project administration—D.F.Q.S. and A.C.; Funding acquisition—J.M.H. and A.C.

## Competing interests

The authors declare no competing interests.
