## [Peer Review File · Communications Biology]

Reviewers' comments:

Reviewer #1 (Remarks to the Author):

It has been publicly known that, in addition to humoral immunities, the hemocyte encapsulation/nodulation and melanization play essential roles in insect innate immunity defense responses against different parasites. In this study, Smith et al. report the employment of different techniques to demonstrate the in situ, in vivo and in vitro melanization processes of the wax moth larvae *Galleria mellonella* or its hemolymph against two mammalian pathogenic fungi. The contribution of melanin to kill the spores of *C. neoformans* is evidenced while *C. albicans* could trigger more intensive melanization responses than *C. neoformans* in *Galleria* but with the immune evasion/escape of melanin/melanization. Overall, the title is not attractive while the obtained data and the collective techniques used/developed in this study may benefit the understanding and future investigation of insect melanization responses against different fungi. Adjustment of conclusion and improvement for clarity are still required.

1. It is confusing that the authors on the one hand proved that the insect melanin is fungicidal on the other however showed that cryptococcal melanin enhances melanization. It would be better to clarify that there are at least five basic types of melanin polymers. Fungi produce either 1,8-dihydroxynaphthalene (DHN) and/or L-DOPA type melanins while insects produce DHI-type melanins etc. that would be structurally different from fungal melanins.

2. Indeed, the insects like *Galleria* have been widely used for the virulence assays of human pathogenic fungi include *C. neoformans* and *C. albicans*. However, the results of this study, at least the first part, confusingly showed that the injected spores would be fully killed by insect melanin/melanization. It would not be true and the quenching of spore GFP signal, i.e., killed by melanization, has not been shown in association with the injected amount of spores and/or insect survival over different amounts of spores. The authors have actually shown that even more intensive melanization could be induced by *C. albicans* than by *C. neoformans* in *Galleria*, the former is still immune evasive.

3. It is somehow overstated that the in vivo and even ex vivo killing of *C. neoformans* spores was solely mediated by insect melanins. In fact, antifungal humoral immunities will be quickly activated in insects to produce antimicrobial peptides (AMPs) and beyond to jointly contribute to the clearance of fungal spores. The experiment design in this study has not ruled out the function of AMPs induced in insects. At least, the point should have been discussed.

4. Fig. 1G, why the control of buffer + PTU would result in fewer CFUs than hemolymph? What can this control tell for?

5. There are quite a lot of repeated Results in Discussion.

6. Fig. 6 only shows the overview of *C. albicans* growth in *Galleria*, which actually cannot summarize the main conclusion of this study, especially the fungicidal effect of melanin against *C. neoformans*.

Minor comments:

Line 16, full name is required for *C. neoformans* first appears here.

Line 130, PI needs a full description.

Lines 320-322, insect pathogenic fungi such as *Beauveria* and *Metarhizium* species actually infect

insects by cuticle penetration with substantial cell wall structure re-modification once enter insect hemocoels, and these fungi evolved the abilities of immune escape from encapsulation/melanization (e.g., *Annu Rev Entomol.* 2017. 62:73-90).

Reviewer #2 (Remarks to the Author):

Summary

The authors present a well written manuscript that explores an aspect of *Galleria mellonella* host response to fungal infection. While *G. mellonella* larvae have proven to be a useful tool in examining fungal infections as well as testing investigational therapeutics, host responses have yet to be fully elucidated and this article addresses an information gap. Real time imaging is used to map the melanization responses to *C. neoformans* and *C. albicans*. The manuscript provides evidence of melanin's role in responding to fungal pathogens and participating in the host immune response. By interrogating two types of fungi, reaction differences can be appreciated between yeast and hyphal forms.

Comments

1. The unit of measure is not clear from figure 1B.
2. During pathogenesis, *C. neoformans* can create a melanin capsule around the cell to protect the fungus from host defenses. Do you distinguish melanin made by *C. neoformans* during pathogenesis from host melanin? Can you see if *C. neoformans* cells have capsules (example: India ink staining)?
3. In figure 1E, both axes are means of the gray values. Should the x-axis be mean of the fluorescent value? Would fungal melanin as opposed to host melanin explain, "Surprisingly, PI staining did not result in the expected fluorescence in nodule-encapsulated yeast cells, but there was staining in some of the hemocytes that surrounded the yeast cells..." Can a *C. neoformans* capsule mutant provide this distinction, for example *cap59*?
4. *C. neoformans* and *C. albicans* can be italicized in the figure legends.
5. Include a rationale statement within the results section for including the *lac1* mutant in figure 2.
6. Why do you think there is a greater melanin response to *C. albicans* compared to *C. neoformans*?
7. In figure 4B, could the string like pattern be associated with the *G. mellonella* fat body? Also, adding an arrow to highlight the pattern would be helpful.
8. What error bars are being referenced in lines 743 and 744?
9. Lines 290-291: "melanin encapsulation" is repeated.
10. In the materials and methods section, line 418, is there an origin reference for the *C. neoformans lac1* mutant?
11. Line 489, change "are" to "were" and line 490 "is" to "was". Also, line 505 should be "analyzed".

Reviewer #3 (Remarks to the Author):

The manuscript by Smith et al entitled "Melanization is an important antifungal defense mechanisms in *Galleria mellonella* hosts" provides some new and careful data on insect immunity against two fungal species (*Cryptococcus neoformans*, *Candida albicans*) detailing the major following findings:

1. Using transgenic GFP expression as a vital marker, the authors describe the strong correlation of melanization to fungal death both in vitro and in vivo against *C. neoformans*.
2. The authors describe the kinetics of the melanization response ex vivo to both *C. neoformans* and *C. albicans*.
3. The study details a clever ex vivo method that the authors used to determine the role of

hemolymph and hemocytes, respectively to the melanization of *C. neoformans* and *C. albicans*.

4. Lastly, using the same *ex vivo* method, the authors detail the melanization kinetics against *C. albicans*, indicating that some yeast cells escape melanization, then switch to hyphal growth, which largely is not melanized, and melanization is activated again by blastoconidial formation.

The manuscript is well written, and provides new insight into the requirements and kinetics of host-microbe interactions in the context of melanization, a fundamental innate immune response in insects. As such it will be of interest to the research community. Below are some minor comments that I hope the authors will consider during revision:

Editorial comments:

1. GFP-based viability assays were described in *Plasmodium berghei* nearly two decades ago by Blandin et al 2004, and deserve a citation.
2. The authors seem to discuss fungal melanin as a trigger (or PAMP) for melanization and the reduced melanization in the *deltalac1* mutant as independent (e.g. Discussion lines 328-343). Is it not likely that *C. neoformans* contains less melanin and thus there is less PAMP to trigger melanization? This should be added to the discussion.
3. One technical limitation of the study is that the *ex vivo* assay only assays hemocytes that are able to attach to surfaces (Lines 491-492). This should at least be mentioned as such in the discussion, as not hemocyte populations are sampled.
4. Line 430: what was the injected dose? Please provide not only the concentration but also the injected volume
5. Line 434: Please provide the anticoagulant buffer composition
6. Line 488: I assume that the second centrifugation was performed under the same conditions as the first, but please detail this.
7. Please provide details on the MatTek dishes that were used in your experiments.

Spelling and Grammar:

1. Line 16: spell out *Cryptococcus*
2. Lines 159-160: please use past tense
3. Lines 293-294: change to "melanin encapsulation of the fungus within nodules"
4. Lines 326: Change "launches" to "launch"
5. Figure legends Figures 3 and 5: please change to past tense throughout.
6. Figure 6: The text in the figure is illegible. Please remove the text from the figure itself and add it to the figure legend.

Reviewers' comments:

Reviewer #1 (Remarks to the Author):

1. **Comment:** It has been publicly known that, in addition to humoral immunities, the hemocyte encapsulation/nodulation and melanization play essential roles in insect innate immunity defense responses against different parasites. In this study, Smith et al. report the employment of different techniques to demonstrate the in situ, in vivo and in vitro melanization processes of the wax moth larvae *Galleria mellonella* or its hemolymph against two mammalian pathogenic fungi. The contribution of melanin to kill the spores of *C. neoformans* is evidenced while *C. albicans* could trigger more intensive melanization responses than *C. neoformans* in *Galleria* but with the immune evasion/escape of melanin/melanization. Overall, the title is not attractive while the obtained data and the collective techniques used/developed in this study may benefit the understanding and future investigation of insect melanization responses against different fungi. Adjustment of conclusion and improvement for clarity are still required.

2. **Response:** We thank the reviewer for the constructive feedback. We have carefully considered all the comments and have extensively revised the manuscript accordingly to improve clarity and the flow of logic. We also better explain how the data in Figure 1, supports a fungicidal role for melanin that contributes in part to the overall fungal death within the host. We agree that the title was too general, and the revised title better reflects the contribution of the work.

3. **Comment:** It is confusing that the authors on the one hand proved that the insect melanin is fungicidal on the other however showed that cryptococcal melanin enhances melanization. It would be better to clarify that there are at least five basic types of melanin polymers. Fungi produce either 1,8-dihydroxynaphthalene (DHN) and/or L-DOPA type melanins while insects produce DHI-type melanins etc. that would be structurally different from fungal melanins.

Response: Thank you for the feedback and pointing out parts of our manuscript that lacked clarity. "Insect melanins are fungicidal" and "fungal melanins elicit the insect melanization response" are not necessarily contradictory or counterintuitive as better explained in the revised manuscript. We believe the fungal melanin is acting like a PAMP or DAMP and activating the melanization response as a consequence, as we better describe in the revised manuscript. The melanins produced by both the insect immune response and *Cryptococcus* are DOPA melanins, and similar structures and properties as they both derive from catecholamine precursors. While other fungi can produce other types of melanin (like DHN melanin), these fungi are not used in this study. When the insect and fungal melanins are first introduced, we have added the clarification that we are referring to exclusively DOPA melanins.

4. **Comment:** 2. Indeed, the insects like *Galleria* have been widely used for the virulence assays of human pathogenic fungi include *C. neoformans* and *C. albicans*. However, the results of this study, at least the first part, confusingly showed that the injected spores would be fully killed by insect melanin/melanization. It would not be true and the quenching of spore GFP signal, i.e., killed by melanization, has not been shown in association with the injected amount of spores and/or insect survivals over different amounts of spores. The

authors have actually shown that even more intensive melanization could be induced by *C. albicans* than by *C. neoformans* in *Galleria*, the former is still immune evasive.

Response: We apologize for not making this clear. In the revision, we further explain that for *C. neoformans*, 6-20% of the melanin-encapsulated fungi survive, not including the fungi freely found within the hemolymph or other tissues, which are sufficient to cause a lethal infection (See response #4 below). For *Candida albicans*, in vitro we find that ~23% of the melanin encapsulated yeast can escape from the encapsulation and continue to grow melanin-evasive hyphae or pseudohyphae. Hence, the melanization reaction is efficiently antifungal, but it is neither the exclusive cause of fungal death, nor does it cause sterilizing immunity, resulting in enough fungal survival to eventually kill the host. *Galleria* survival increases with smaller inoculum of *C. neoformans*, to the point where, presumably, the melanin-based immune response combined with the other aspects of insect immunity are able to completely kill/protect the larva from the fungus (Smith, D. F., & Casadevall, A. (2022). On the relationship between Pathogenic Potential and Infective Inoculum. *PLoS Pathogens*, 18(6), e1010484).

5. **Comment:** 3. It is somehow overstated that the in vivo and even ex vivo killing of *C. neoformans* spores was solely mediated by insect melanins. In fact, antifungal humoral immunities will be quickly activated in insects to produce antimicrobial peptides (AMPs) and beyond to jointly contribute to the clearance fungal spores. The experiment design in this study has not ruled out the function of AMPs induced in insects. At least, the point should have been discussed.

Response: Thank you for pointing this out. As some of the non-melanin encapsulated fungi are dead within the nodules, we did not intend to imply that melanization is the sole cause of fungal death within the nodules. We have included the following sentences in the results section to help clarify the role of melanin in fungal death.

“The non-melanin encapsulated fungi had a survival rate ranging from ~22, 33, 59, and 38% for the room temperature 24 and 72 h and the 30°C at 24 and 72 h conditions, respectively. The melanin-encapsulated fungi had a survival rate of ~6, 10, 21, and 16% for the room temperature 24 and 72 h and the 30°C at 24 and 72 h conditions, respectively (Figure 1C and 1D). Melanin-encapsulation thus corresponds to a ~21, 34, 92, and 35% greater amount of fungal death, respectively. This indicates that melanization contributes to a significant, but not exclusive, amount of fungal death during infection. Additional causes of fungal death could be antimicrobial peptides and respiratory bursts.” We have revised Figure 1D to demonstrate the relationship between melanin encapsulation and fungal survival more clearly.

6. **Comment:** 4. Fig. 1G, why the control of buffer + PTU would result in fewer CFUs than hemolymph? What can this control tell for?

Response: Since the buffer consists of mostly salts and only 1 mM of the sodium bicarbonate carbon source, we would not expect the fungus to grow in the buffer alone, and as a result, has low CFUs, whereas in the hemolymph conditions, there are nutrients that the fungus can use to grow and replicate. This buffer alone with PTU control tells us 1) that the PTU itself was not fungicidal, fungistatic, nor enhanced the growth of the fungus on its own, and 2) that the wells with high degrees of melanin have fungal growth comparable to the buffer alone, indicating that the melanization is controlling the growth of the fungus, whereas the fungus could grow within the melanin-inhibited hemolymph. We have added the following sentence to this paragraph, “The buffer with PTU control without hemolymph

shows us that the PTU itself is non-toxic to the fungus nor does it account for increased fungal growth in the high PTU concentrations.”

7. **Comment:** 5. There are quite a lot of repeated Results in Discussion.
Response: We have removed some “discussion”-like sentences from the results section, but we have kept most of the discussion section as is, as we discuss our results, the implications of our results, and the strengths and limitations of the methods we have used and/or developed.
8. **Comment:** 6. Fig. 6 only shows the overview of *C. albicans* growth in *Galleria*, which actually cannot summarize the main conclusion of this study, especially the fungicidal effect of melanin against *C. neoformans*.
Response: For this figure, we were only intending to summarize the *Candida albicans* findings, as it was introduced in the summary paragraph for the *C. albicans* findings, as we believe the illustration helps combine that data into a story. In the revision, we have included a summary figure of the GFP viability protocol as part of Supplementary Figure 4.

Minor comments:

9. **Comment:** Line 16, full name is required for *C. neoformans* first appears here.
Response: We have made this correction.
10. **Comment:** Line 130, PI needs a full description.
Response: We have included “propidium iodide” when introducing the compound for the first time and the abbreviation PI.
11. **Comment:** Lines 320-322, insect pathogenic fungi such as *Beauveria* and *Metarhizium* species actually infect insects by cuticle penetration with substantial cell wall structure re-modification once enter insect hemocoels, and these fungi evolved the abilities of immune escape from encapsulation/melanization (e.g., *Annu Rev Entomol.* 2017. 62:73-90).
Response: Thank you for this point, which we cite in the revision. We believe our findings showing how fungal melanins activate the insect immune response hint at a possible/additional way in which entomopathogenic fungi have evolved to not produce melanin in order to evade the melanization response during infection.

Reviewer #2 (Remarks to the Author):

Summary

12. **Comment:** The authors present a well written manuscript that explores an aspect of *Galleria mellonella* host response to fungal infection. While *G. mellonella* larvae have proven to be a useful tool in examining fungal infections as well as testing investigational therapeutics, host responses have yet to be fully elucidated and this article addresses an information gap. Real time imaging is used to map the melanization responses to *C. neoformans* and *C. albicans*. The manuscript provides evidence of melanin’s role in responding to fungal pathogens and participating in the host immune response. By interrogating two types of fungi, reaction differences can be appreciated between yeast and hyphal forms.

Response: We greatly appreciate the reviewer's summary of our work, and the helpful constructive feedback provided on the manuscript below.

Comments

13. **Comment:** The unit of measure is not clear from figure 1B.

Response: We have edited Figure 1B to include the unit, which is the percent frequency of each of the fluorescence statuses.

14. **Comment:** 2. During pathogenesis, *C. neoformans* can create a melanin capsule around the cell to protect the fungus from host defenses. Do you distinguish melanin made by *C. neoformans* during pathogenesis from host melanin? Can you see if *C. neoformans* cells have capsules (example: India ink staining)?

Response: *C. neoformans* does produce melanin during infection in *Galleria*, but we do not differentiate it from the melanin produced by the insect other than through appearance. The melanin produced by *C. neoformans* is located on the innermost layer of the cell wall, while the melanin produced by the insect immune response is located primarily outside of this area (within and surrounding the polysaccharide capsule of *C. neoformans*). For clarification, cryptococcal melanins are produced in the fungal cell wall, the cryptococcal capsule is composed of polysaccharide and surrounds the cell wall (and is mostly visible within the nodules), and the term for the process by which insects surround microbes with melanin and hemocytes/nodules is called "encapsulation." We have attached a diagram below to help with the clarification. In the revised manuscript we have included this diagram as part of Figure 1, and replaced the existing figure 1A.

15. **Comment:** 3. In figure 1E, both axes are means of the gray values. Should the x-axis be mean of the fluorescent value?

Response: Since the images are in black and white, technically the readout is the mean gray value. To avoid confusion, we have retitled the axes to read "mean pixel value" instead of "mean gray value."

16. **Comment:** Would fungal melanin as opposed to host melanin explain, "Surprisingly, PI staining did not result in the expected fluorescence in nodule-encapsulated yeast cells, but there was staining in some of the hemocytes that surrounded the yeast cells..." Can a *C. neoformans* capsule mutant provide this distinction, for example cap59?

Response: That is a very interesting point – whether the hemocyte death is being caused by virulence factors in the fungus or through normal death/lysis that hemocytes undergo within nodules as part of the immune response. We are not sure the acapsular mutant would answer this question, as the role of capsule is mostly to protect the fungus and avoid

phagocytosis/immune activation, and thus would likely not be cytotoxic.

17. **Comment:** 4. *C. neoformans* and *C. albicans* can be italicized in the figure legends.

Response: We have made the edits to italicize *C. albicans* and *C. neoformans* in the figure legends.

18. **Comment:** 5. Include a rationale statement within the results section for including the *lac1* mutant in figure 2.

Response: We have included a few sentences introducing laccase and why we did these experiments. This reads, "In *C. neoformans*, the laccase enzyme Lac1 is a major virulence factor, as it is responsible for the oxidation of catecholamines, and thus responsible for melanization (ref). It has also been suggested that laccase plays roles in virulence beyond melanization within mammalian hosts (ref). In *G. mellonella*, *lac1* Δ mutants had attenuated virulence (ref).

19. **Comment:** 6. Why do you think there is a greater melanin response to *C. albicans* compared to *C. neoformans*?

Response: We have not performed any experiments yet to determine why *C. albicans* elicits a more robust melanization response. We think that the *C. albicans* Beta-glucan and/or mannan layers trigger the melanization response, based on our understanding of the cell wall structures between these species, and of common PAMPs recognized by insect immune response. In contrast, in *C. neoformans* the cell is shielded by the capsule which may prevent direct contact between hemolymph and cell wall components. A similar situation occurs in the mammalian response where Dectin interacts with cell wall in *C. albicans* and has an important role in immune recognition but not for *C. neoformans*. We added a sentence to note this possible parallel with references in the discussion section.

20. **Comment:** 7. In figure 4B, could the string like pattern be associated with the *G. mellonella* fat body? Also, adding an arrow to highlight the pattern would be helpful.

Response: Thank you for the suggestion, and we have added arrows accordingly to 4C and 4D. That is a good question. We do not believe the pattern is due to association or particular tropism to the fat body of the larvae. The *G. mellonella* fat body is found throughout the larvae and has a more amorphous structure. In addition, when we perform tissue dissections of infected larvae, we don't see association of the nodules with the fat body besides being found in the "nooks and crannies" of the tissue (Supplementary Figure 2).

21. **Comment:** 8. What error bars are being referenced in lines 743 and 744?

Response: Thank you for pointing this out, we meant to write "scale bars," instead of error bars. It has since been corrected.

22. **Comment:** 9. Lines 290-291: "melanin encapsulation" is repeated.

Response: The repeated "melanin encapsulation" has been removed.

23. **Comment:** 10. In the materials and methods section, line 418, is there an origin reference for the *C. neoformans lac1* mutant?

Response: The origin of the *lac1* Δ strain is the mutant library made by Dr. Jennifer Lodge obtained through the Fungal Genetics Stock Center. We have updated the material and methods section to reflect this information.

24. **Comment:** 11. Line 489, change "are" to "were" and line 490 "is" to "was". Also, line 505 should be "analyzed".

Response: The suggested changes have been made.

Reviewer #3 (Remarks to the Author):

25. **Comment:** The manuscript by Smith et al entitled " Melanization is an important antifungal defense mechanisms in Galleria mellonella hosts" provides some new and careful data on insect immunity against two fungal species (Cryptococcus neoformans, Candida albicans) detailing the major following findings:

1. Using transgenic GFP expression as a vital marker, the authors describe the strong correlation of melanization to fungal death both in vitro and in vivo against C. neoformans.
2. The authors describe the kinetics of the melanization response ex vivo to both C. neoformans and C. albicans.
3. The study details a clever ex vivo method that the authors used to determine the role of hemolymph and hemocytes, respectively to the melanization of C. neoformans and C. albicans.
4. Lastly, using the same ex vivo method, the authors detail the melanization kinetics against C. albicans, indicating that some yeast cells escape melanization, then switch to hyphal growth, which largely is not melanized, and melanization is activated again by blastoconidial formation.

The manuscript is well written, and provides new insight into the requirements and kinetics of host-microbe interactions in the context of melanization, a fundamental innate immune response in insects. As such it will be of interest to the research community. Below are some minor comments that I hope the authors will consider during revision:

Response: We greatly appreciate Reviewer 3's summary of our manuscript, and their support of our work. We also want to thank the reviewer for their helpful feedback, comments, and edits below. We have tried to address these suggestions to the best of our ability.

Editorial comments:

26. **Comment:** GFP-based viability assays were described in Plasmodium berghei nearly two decades ago by Blandin et al 2004 and deserve a citation.

Response: We agree. The reference has been added in the first results section paragraph, which reads, "A similar assay using GFP signal as a proxy for studying parasite viability within mosquitoes has been previously described (ref)."

27. **Comment:** The authors seem to discuss fungal melanin as a trigger (or PAMP) for melanization and the reduced melanization in the deltalac1 mutant as independent (e.g. Discussion lines 328-343). Is it not likely that C. neoformans contains less melanin and thus there is less PAMP to trigger melanization? This should be added to the discussion.

Response: Thank you very much for this insight. We missed this interpretation of our results and have added it to our discussion section. We had been focused on interpreting our results in the context of previously reported roles of laccase in interactions with the melanin-based immune response, which has been reported in fungi that do not produce melanin.

28. **Comment:** 3. One technical limitation of the study is that the ex vivo assay only assays hemocytes that are able to attach to surfaces (Lines 491-492). This should at least be mentioned as such in the discussion, as not hemocyte populations are sampled.

Response: Thank you for pointing this out. While we have assumed the hemocyte population is representative, we have added the following sentence to the discussion, "Since we only used hemocytes that adhere to the coverslip, we are making an assumption that the hemocyte population is representative, while there might be a predilection for the adhesion of some hemocyte subtypes compared to others."

29. **Comment:** 4. Line 430: what was the injected dose? Please provide not only the concentration but also the injected volume
Response: The information on inoculum is described the preceding section, but we have now added the information of volume (10 μ l) and infectious inoculum (10^5 /larva) to this part of the methods section.
30. **Comment:** 5. Line 434: Please provide the anticoagulant buffer composition
Response: The recipe for the anticoagulation buffer has been added.
31. **Comment:** Line 488: I assume that the second centrifugation was performed under the same conditions as the first, but please detail this.
Response: The information for the second centrifugation was added.
32. **Comment:** Please provide details on the MatTek dishes that were used in your experiments.
Response: Product details have been now included.
- Spelling and Grammar:
33. **Comment:** 1. Line 16: spell out Cryptococcus
Response: Correction was made
34. **Comment:** 2. Lines 159-160: please use past tense
Response: Correction was made
35. **Comment:** 3. Lines 293-294: change to "melanin encapsulation of the fungus within nodules"
Response: Correction was made as suggested.
36. **Comment:** 4. Lines 326: Change "launches" to "launch"
Response: Correction made as suggested
37. **Comment:** 5. Figure legends Figures 3 and 5: please change to past tense throughout.
Response: Corrections made to the figure legends as suggested.
38. **Comment:** Figure 6: The text in the figure is illegible. Please remove the text from the figure itself and add it to the figure legend.
Response: The figure has been changed, and the text has been removed and placed into the figure legend.

REVIEWERS' COMMENTS:

Reviewer #1 (Remarks to the Author):

The manuscript has been substantially improved after revisions, and my concerns have been satisfactorily addressed.

Reviewer #2 (Remarks to the Author):

Prior comments, suggestions, and questions have been adequately addressed. The authors provide a well-written article that explores an aspect of *Galleria mellonella* host response to fungal infection. While *G. mellonella* larvae have proven to be a useful tool in examining fungal infections as well as testing investigational therapeutics, host responses have yet to be fully elucidated and this article addresses an information gap. The manuscript provides evidence of melanin's role in responding to fungal pathogens and participating in the host immune response. By interrogating two types of fungi, reaction differences can be appreciated between yeast and hyphal forms.

A few minor editorial items need to be corrected:

1. Line 431: "at the same..."
2. Line 419: "important role..."
3. Line 423: Change "of the rate..." to "the rate of..."
4. There appears to be an editing note in line 696, "in which condition?"